# Application of Geophysical and Hydrogeochemical Methods to the Protection of Drinking Groundwater in Karst Regions

**DOI:** 10.3390/ijerph17103627

**Published:** 2020-05-21

**Authors:** Kai Song, Guangxu Yang, Fei Wang, Jian Liu, Dan Liu

**Affiliations:** Faculty of Geosciences and Environmental Engineering, Southwest Jiaotong University, Chengdu 610031, China; songkailw@163.com (K.S.); yangguangxu@my.swjtu.edu.cn (G.Y.); jlau@swjtu.edu.cn (J.L.); dliu@swjtu.edu.cn (D.L.)

**Keywords:** karst conduit systems, drinking groundwater source, hydrochemical process, water–rock interaction

## Abstract

To provide theoretical support for the protection of centralized drinking groundwater sources in karst areas, it is necessary to accurately identify the development of karst conduits and analyze the differences in hydrogeochemical characteristics of different karst systems. This provides a scientific basis for the accurate designation of risk zones that may cause drinking groundwater pollution. In this study, a geophysical survey, hydrogeological chemical process analysis and optimized fuzzy cluster analysis were used to gradually improve the understanding of karst water systems. AMT and HDR methods were used to calibrate the resistivity around the water-filling karst conduits, which ranged from 39 to 100 Ω·m. A total of seven karst systems were identified, including four karst systems in the north of the study area, one karst system in the west and two karst systems in the south. Analysis of the hydrochemical data showed that HCO_3_-Ca and HCO_3_-Mg-Ca types accounted for 90% of all samples. The δD and δ^18^O values of their main conduits were −51.70‰ to −38.30‰ and −7.99‰ to −5.96‰, respectively. The optimized fuzzy clustering analysis method based on the weight of variables assigned by AHP more accurately verified karst water systems. Based on these findings, the drinking groundwater source risk zone was designated with an area of 33.90 km^2^, accounting for 34.5% of the study area. This study effectively improved the rationality and accuracy of the designation of drinking groundwater source risk zones in karst areas, and provided a scientific basis for the identification of karst water systems and decision-making of drinking groundwater source protection in karst areas.

## 1. Introduction

Low temperature groundwater in karst development areas, which is mainly discharged in the form of karst springs or underground rivers, has become an important water source because of its stable quality and abundance [1,2,3]. However, the high heterogeneity of carbonate rocks in karst areas greatly increases the difficulty of pollution prevention and control [4]. Additionally, the strong hydraulic alternation conditions make this kind of water more susceptible to pollution and rapid migration, thereby posing health risks to the supply objects [5,6]. In view of the concealment of groundwater contamination and difficulty in remediation [7], prevention measures should be taken as the main means for groundwater protection when conducting industrial development, especially in karst areas [8,9,10]. Studies of heterogeneity, development degree and hydrogeochemical characteristics in typical karst aquifers can provide important information for sustainable development of groundwater and safety management of drinking groundwater sources [11,12], which is particularly important for developing the complex karst conduit systems and being the only drinking groundwater source for more than 15,000 people in the study area of this paper [13].

The geophysical survey is a qualitative or semi-quantitative exploration method based on obtaining differential information pertaining to electrical conductivity, electromagnetism or energy transmission, which is widely used studies pertaining to development and hydrogeological characteristics of a karst system [14,15]. McCormack et al. [16] overcame various difficulties in the application of traditional karst investigation techniques (such as the tracer method and the hydrogeochemistry analysis method) caused by a complex mixture of upland, lowland and coastal karst by using a high density electrical method, which is a common method for the construction of hydrogeological conceptual models, to identify the hydrogeological characteristics of a typical karst area. When compared with the high density resistivity method (HDR), which is suitable for media with a depth of less than 200 m [17,18], the audio frequency magnetotellurics method (AMT) can obtain resistivity information regarding deep media [19]. Wang et al. [20] used this method to successfully identify and forecast concealed karst conduits along a typical tunnel.

Hydrogeochemical methods, which investigate the distribution, migration and dispersion of chemical elements and isotopes in groundwater to interpret geochemical processes, are mainly applied to solve practical problems such as identification of groundwater sources and division of hydrogeological units [21,22]. Previous studies have shown that solutes reflect the hydrogeochemical background conditions of a study area, and are influenced by both natural and human activities [23]. Zhu et al., Sharma et al. and Chetelat et al. used multiple ion proportionality and chemical equilibrium analytical methods to analyze the relevant information pertaining to groundwater affected by precipitation input, the water–rock interaction, anthropogenic activity and other factors [24,25,26]. Based on analysis of the changes in characteristics of hydrogeochemical characteristics combined with isotopic information obtained along the groundwater runoff route and relative statistical methods, Leybourne et al. [27] and Yuan et al. [28] successfully interpreted the origin of groundwater and the dominant hydrogeochemical processes controlling groundwater water quality in a karst area.

Thus, this study was conducted to accurately identify karst conduits and effectively protect karst water systems that serve as drinking groundwater sources in a karst development area. AMT and HDR were used to identify the development of karst conduits, and groundwater samples were subjected to hydrogeochemical and isotopic analysis to identify the ionic component characteristics and hydrogeochemical process differences of groundwater in different karst systems. The optimized fuzzy cluster analysis method with weights assigned by variable indicators was then used to classify karst systems. The suitability of different karst systems in the area to serve as drinking groundwater sources was comprehensively studied and appropriate risk zones were designated according to the suitability of industrial activities.

## 2. Location and Hydrogeology of the Study Area

The study area was located in the hilly areas of the southeastern margin of the Sichuan Basin. It was defined based on hydrogeological conditions and karst development, corresponded to the recharge area of the Gu Song River upper reaches and had an area of 101.68 km^2^. Based on historical rainfall data, the rainfall is obviously controlled by seasonal factors, with 72% of the annual rainfall occurring from May–September, and only 8.8% occurring during winter.

According to the hydrogeology survey, karst depressions and sinkholes are densely distributed in the study area, and surface water systems are rarely developed. The only drainage datum plane, the Gu Song River, runs off from west to east across the study area. The geomorphology is obviously controlled by water erosion and accumulation as follows: the surrounding areas belong to the landform of a medium incision mountain in the shape of a spire. The corresponding emergence stratum is clastic rock strata of the Feixianguan and Xujiahe formation of the Triassic, with an exposed area of 30.84 km^2^, accounting for 30% of the study area. The emergence stratum in the center of the study area is carbonate of the Leikoupo and Jialinjiang formations of the Triassic, which have formed low mountains of hoodoos and peak clusters with an area of 63.44 km^2^, accounting for about 63% of the study area. Along the Gu Song River, the eluvial clay and alluvial-proluvial sandy gravel and pebbles are deposited to form banded karst valleys, with an area of 7.40 km^2^, accounting for about 7% of the study area (Figure 1).

Geological investigation of the karst development showed that the horizontal solution fissures in carbonate rocks were mainly developed along rock stratification, while the vertical solution fissures were mainly controlled by tectonic activities and vertical erosion of precipitation, and the maximum width of the fissure exceeded 1 m. Part of the solution fissures gradually eroded and developed into karst conduits or caves. The identified caves were 3–5 m high and 5–10 m wide, with a maximum length of several kilometers. According to data from 71 groups of karst springs observed in multiple field investigations in this study, the minimum mean flow and maximum mean flow of all springs were 0.1 and 1215.0 L/s respectively. The runoff process of groundwater in the study area was characterized by repeated alternation of surface runoff and underground runoff. It was not feasible or necessary to divide the boundary of each spring drainage area. Therefore, this paper attempted to identify the main karst conduits by geophysical survey, and then conducted further hydrogeochemical study based on the identification results.

## 3. Materials and Methods

### 3.1. Geophysical Method

Based on the surface elevation and buried depth of the saturated zone of the karst aquifer, HDR and AMT were used to investigate the development of buried karst. In combination with the surface karst investigation, the developments of karst conduits in the study area were preliminarily identified. HDR adopted the Wenner four-electrode method of the electrical measurement system, with a current resolution of 0.01 μA and a current accuracy of 0.1%. Swedish software RES2DINV was applied to conduct the inversion of two-dimensional resistivity of the detection results. After topographic correction and removal of the discontinuity point, repeated parameter adjustments and iterative calculations were conducted until the root mean square error (RMSE) between the calculated resistivity and the measured resistivity was in the range of 3.6–8.79%. The reliable 2-D inversion resistivity profiles of HDR were obtained and used to interpret the buried depth of karst conduits. AMT adopted the MTU-5A GPS satellite synchronization magnetotelluric instrument produced by the Phoenix Company (Canada) and used its own AMT data interpretation software package for the inversion to obtain the electrical structure of underground media.

According to the hydrogeological investigation of surface karst and the application conditions of geophysical survey, HDR was mainly adopted in the middle of the study area with low altitude and shallow karst development depth. There were 20 survey lines (HDR-1–HDR-20), the distance between survey points was 5 or 7.5 m and the total length of the survey lines was 53.14 km, including 9435 survey points. AMT was mainly adopted south and northwest of the study area, where there was high altitude and deep karst development depth. There were six survey lines (AMT-1–AMT-6), the distance between survey points was 40 m and the total length of the survey lines was 25.84 km, including 684 survey points (Figure 1).

### 3.2. Sampling and Laboratory Analysis

According to the development of karsts obtained by geophysical survey and the numerous groundwater outcrops distribution identified by surface hydrogeological survey, this study conducted optimal sampling based on the principle that the samples analysis results need to basically reflect the chemical characteristics of groundwater in the processes of recharge, runoff and discharge. As shown in Figure 1, 26 points, including springs, underground river outlets and water filled caves were selected as groundwater sampling points, and 4 surface water sections were selected as surface water sampling points. The selected sampling points of groundwater and surface water were sampled once in dry season (November 2018) and wet season (August 2019) respectively. In addition, a rain sample were collected in wet season. The collection, storage and monitoring of samples were strictly in accordance with the relevant standards of the American Public Health Association [29] and China [30,31]. Physical, chemical, metal, organic and inorganic factors were monitored. Table 1 shows the detailed information of each sampling point, and Table 2 shows the monitoring method and limit for each factor.

To further analyze the hydrogeochemical process and characteristics of the karst system in the study area, stable isotopes δ^18^O and δD were added as monitoring factors during the wet season. The δ^18^O and δD were measured using a stable isotope mass-spectrometer at the Key Laboratory of Karst Dynamics, Ministry of Natural Resources. The isotopic composition of oxygen was determined through the water-CO_2_ equilibration technique and that of hydrogen through the water H_2_ equilibration technique using a platinum catalyst. The reproducibility calculated from standards systematically interspersed in the analytical batches was ±0.2‰ for δ^18^O and ±1.5‰ for δD. The laboratory standards were regularly calibrated according to international standards (Vienna Standard Mean Ocean Water, VSMOW). The isotope ratios are reported in the denotation:(1)$$\delta_{sample}\left(‰\right)=\left[\left(R_{sample}/R_{standard}\right)-1\right]\times 10^{3}$$
where R is the molar concentration ratio of D to H or ^18^O to ^16^O. δ^18^O and δD were reported relative to VSMOW [32].

## 4. Result

### 4.1. Karst Characteristics and Identification of Karst Conduits

#### 4.1.1. Calibration of Medium Resistivity

Controlled by the differences in mineral composition, porosity and groundwater occurrence, there are obvious differences in the resistivity between different rocks [33,34]. Therefore, to ensure accuracy of the identification, it was necessary to calibrate the resistivity of different types of rocks before conducting the geophysical survey in the study area to provide the basis for interpretation of karst conduits. HDR and AMT identification results showed that the resistivity of clastic rock with a buried depth of less than 40 m was between 10^2^–10^3^ Ω·m because of the existence of the shallow weathered fracture aquifer, but the resistivity increased to more than 10^3^ Ω·m as the buried depth increased.

In the carbonatite rocks area, the resistivity was between 251 and 1000 Ω·m from the vadose zone to the epiphreatic zone. Under the epiphreatic zone, the weak karst development area had a resistivity of more than 10^3^ Ω·m, which was similar to the characteristics of clastic rock. Geophysical survey were carried out for several water-filled karst conduits identified by field hydrogeological investigation. The result showed that the resistivity of water-filled karst conduits was between 39 and 100 Ω·m. The unique conductivity of karst conduits, which was less than one order of magnitude than other identification areas, indicated the high heterogeneity of the carbonate area and ensured the accuracy of karst conduits identification.

#### 4.1.2. Identification Results

Based on the results of resistivity calibration and considering the continuity of geophysical survey profiles and surface karst development, karst conduit 1–15 were identified in the study area (Figure 1). Taking the Gu Song River as a reference point, karst conduit 1–6 were identified in the north, encompassing four underground river systems. Each underground river system in the study area is named after its underground river outlet and the names of the four underground river systems are Xin Zhai Spring (XZS), Long Xian Spring (LXS), Giant Salamander Spring (GSS) and Black Cave Spring (BCS). Karst conduit 7–10 were identified in the west of the Gu Song River, where the groundwater ran and drained through the Fish Well Spring (FWS). Karst conduit 11–15 were identified in the south, where there were two underground river outlets, Big Fish Spring (BFS) and Small Fish Spring (SFS). The detailed hydrogeological parameters of each karst conduit are shown in Table 3.

The identification data in Table 3 shows that the development of karst conduit 1–6, which were included by XZS, LXS, GSS and BCS in the north, were similar, with an initial elevation of 851–967 m, outlet elevation of 454–514m, a possible main conduit length of 1.69–3.40 km and an possible average hydraulic gradient of 0.12–0.19. Karst conduit 14 and 15 included by SFS had initial elevations of 636–840 m, an outlet elevation of 459 m, the possible main conduit length of 5.73 km and an possible average hydraulic gradient of 0.07–0.04. It is worth noting that FWS and BFS are the centralized urban drinking water sources in the study area. The FWS, which is the drinking groundwater source for 3000 people, has a mean flow of 325.3 L/s during dry season and 813.2 L/s in the wet season, and karst conduit 7–10 within the FWS developing from the northwest to the southeast of the Gu Song River. The longest karst conduit in the area, 9, is possibly 7.21 km. The interpretation information of karst conduit 9 shows that the initial elevation is 1221 m, the buried depth of runoff area is 63–139 m and the elevation of the outlet is 528 m, as shown in Figure 2a. The BFS is the drinking groundwater source for 12,000 people, with a mean flow of 529.3 L/s in the dry season and 1215.0 L/s in the wet season. This area included karst conduit 11–13 developing from south to north. The longest conduit in the area 12 is possibly 7.21 km. The interpretation information for karst conduit (12) showed that the initial elevation was 887 m, the buried depth of the runoff area was 50–181 m and the elevation of the outlet was 475 m, as shown in Figure 2b. According to the function, FWS and BFS have become the focus of groundwater environmental protection in the study area.

### 4.2. Basic Characteristics of Groundwater Quality and Chemistry

The collected water samples included rain, groundwater and surface water. The pH of rain was 7.56, the total hardness was 8.51 mg/L and the total dissolved solids content was 34.9 mg/L. The chemical types of groundwater and surface water were similar. According to data of groundwater (Table 4) and surface water collected during the dry and wet season, the pH value was between 7.02–8.46 and 7.15–8.25, respectively; the total hardness was 55–240 and 30–330 mg/L; and the total dissolved solids were between 71–276 and 71–504 mg/L. Na^+^, K^+^, Ca^2+^, Mg^2+^, Cl^−^, SO_4_^2−^, HCO_3_^−^ and other factors accounted for more than 95% of the total dissolved solids in water samples. At the same time, all hydrochemical data had passed the ionic charge balance test of anions and cations, indicating that the measured data are reasonable and credible. According to the classification of Shoka Lev [35,36], the hydrochemical type of rainwater is HCO_3_-SO_4_-Na-Ca, while the chemical types of surface water and groundwater were similar, and mainly HCO_3_-Ca and HCO_3_-Mg-Ca, which accounted for 90% of the collected water samples. The low salinity and single hydrochemical type indicate that the circulation and alternation of groundwater are intensive and the rock type involved in the water rock interaction are not complex.

In addition to the above factors, only Fe, Cd and Al were detected in the 10 metal factors, and the detection rate and concentration value did not fluctuate significantly during the dry and wet season. According to the data from the dry season, the detection rate and value of Fe were 56.7% and 3.1 × 10^−2^–6.8 × 10^−2^ mg/L, respectively, while the detection rate and value of Cd were 46.7% and 1.9 × 10^−5^–1.0 × 10^−4^mg/L. Additionally, Al was detected in all water samples with a detection value of 8.0 × 10^−3^–9.8 × 10^−2^ mg/L. No volatile phenols or petroleum were detected, and COD_Mn_ value was 0.75–1.67. Three kinds of nitrogen compounds were detected in water samples, the detection rate of NO_3_^−^ was 100% and the detection value was 3.11–14.37 mg/L during dry season and wet season. NO_2_^−^ was only detected during dry season, with a detection rate of 6.7% and an average value of 0.27 mg/L. The detection values of NH_4_^+^ were 46.7% and 60% in dry season and wet season, respectively, and the concentration was 0.017–0.19 mg/L. The concentration of nitrogen compounds in groundwater was higher during the wet season than the dry season, which may have been because of differences in the groundwater recharge intensity after rainfall leaching through surface materials between the dry season and wet season. Additionally, the strong hydraulic alternation conditions also increased the oxygen content in groundwater, resulting in the main nitrous oxide in groundwater being NO_3_^−^ and almost no NO_2_^−^ being detected. In general, 26 factors monitored in this study meet the drinking groundwater source standard of groundwater [37]. The concentrations of main components in groundwater were relatively low, especially for Cl^−^, SO_4_^2−^ and NO_3_^−^, indicating that the input of exogenous substances caused by human activities is not the controlling factor of groundwater quality, while natural factors mainly control the solute composition and evolution of groundwater. Therefore, the primary measure of drinking groundwater source protection in the study area is pollution prevention and control of the groundwater environment.

### 4.3. Stable Oxygen and Hydrogen Isotopes

This section analyzes the differences between groundwater recharge sources based on hydrogen and oxygen stable isotopes. The δD and δ^18^O values of groundwater in the study area ranged from −51.70‰ to −27.60‰ and −7.99‰ to −5.15‰, with average values of −40.60‰ and −6.74‰, respectively. The values of δD and δ^18^O of the surface water ranged from −41.60‰ to −37.90‰ and −7.04‰ to −6.82‰, with average values of −40.10‰ and −6.92‰, respectively. The values of δD and δ^18^O of rain were −23.40‰ and −4.47‰.

The differences in δ D and δ^18^O values were closely related to the water circulation process of the recharge source [38,39]. All spots of δ^18^O vs. δD of the karst water and surface water in the study area were located near the global metric water line (GMWL) [40], indicating that precipitation is the main recharge source of groundwater and surface water in the area (Figure 3). It is not surprising that the distributions of δD and δ^18^O in the surface water and groundwater samples were similar, because they represent the strong hydraulic conductivity among precipitation, surface water and karst aquifer. Using the least square regression equation method to fit the δ^18^O vs. δD relationship of all water samples yielded δD = 7.33δ^18^O + 9.14. The slope of 7.33 was close to that of the GMWL equation δD = 8δ^18^O + 10, and the intercept of 9.14 was also close to that of the GMWL equation, indicating that δ D and δ^18^O are not affected by the secondary evaporation process in the study area.

The distribution of the spots of δ^18^O vs. δD in Figure 3 reveals the obvious difference between karst systems in the north and the FWS, BFS and SFS. The δD and δ^18^O values of the eight groups of water samples in the north karst system were between −43.50‰ and −36.20‰ and −7.23‰ and −6.12‰, respectively. As shown in Figure 3, the distribution of the spots was relatively more concentrated, indicating that recharge and runoff areas are relatively small. The δD and δ^18^O values of the water samples of FWS, BFS and SF were between −51.70‰ and −38.30‰ and −7.99‰ and −5.96‰, respectively, and the distribution range of δ D and δ^18^O values was obviously larger. In addition, there is a type of karst spring with small flow and typical characteristics of in-situ recharge and discharge, which has δD and δ^18^O values close to those of rain. Because of the rapid circulation of groundwater in karst voids, the water rock interaction between the groundwater and surrounding rock is weak. Sample GW5 is this type and the corresponding calcite saturation index (SIC) and dolomite saturation index (SID)are the minima of all samples, which were −1.551‰ and −4.075‰, respectively.

## 5. Discussion

### 5.1. Analysis of Hydrochemical Processes Affecting Groundwater Solute Components

Based on the hydrogeological conditions and the development of karsts obtained by geophysical prospecting, this study conducted optimal sampling and analysis of groundwater to further analyze the development of karst conduits by applying the chemical equilibrium analytical method and the multiple ion proportionality method.

#### 5.1.1. Source Analysis of Cl^−^ and Na^+^

Cl^−^ and Na^+^ in groundwater are mainly derived from the input of precipitation and human agricultural activities. The molar ratio of Cl^−^ to Na^−^ is usually used to analyze the sources of Cl^−^ and Na^+^ in the study area [41]. The molar ratio of Cl^−^/Na^+^ in the atmospheric precipitation input from the ocean is 1.16 [42]; therefore, the ratio of Cl^−^/Na^+^ can be used to determine whether Cl^−^ and Na^+^ in the study area is from the atmospheric precipitation input. As shown in Figure 4, Cl^−^/Na^+^ ranged from 0.03 to 1.36, with an average value of 0.58. The Cl^−^/Na^+^ ratio of some water samples was close to 1:1.16, indicating that these water samples are heavily influenced by sea salt deposition. However, 90% of the samples had a molar ratio of Cl^−^/Na^+^ less than 1.16, indicating that there are additional Na^+^ ions involved in the chemical reactions in the study area.

Carbonate rock strata were mainly distributed in the study area, while Triassic clastic rock strata were distributed in the hilly area on the outer edge of the study area, and the mineral compositions of the rocks distributed in the study area were mainly carbonate and silicate. No unique stratum containing Na^+^ minerals was found in the study area. Therefore, the weathering and dissolution of carbonate and silicate strata were the main sources of Ca^2+^, Mg^2+^, Na^+^ and K^+^. If it is assumed that exogenous acids such as sulfide oxidation are only used to balance Ca^2+^ and Mg^2+^ in the water, it can be inferred that [Ca^2+^ + Mg^2+^]* = [Ca^2+^ + Mg^2+^]-[SO_4_^2−^ + NO_3_^−^] based on the dissolution of carbonate and silicate rocks. The ratio of [Ca^2+^ + Mg^2+^]*/[HCO_3_^−^] represents the relative content of carbonate. The closer the ratio is to 1, the more dominant the dissolution of carbonate is in groundwater. Similarly, [Na^+^ + K^+^]* = [Na^+^ + K^+^]-[Cl^−^] comes from the weathering of silicate [43]. Therefore, the ratios of [Ca^2+^ + Mg^2+^]*/[HCO_3_^−^] and [Na^+^ + K^+^]*/[HCO_3_^−^] in water reflect the relative control degrees of carbonate and silicate rocks, respectively (Figure 5). From section D4 to D1, surface water flows from upstream to downstream, and gradually flows to the hinterland of the karst development area. The corresponding ratio of [Ca^2+^ + Mg^2+^]*/[HCO_3_^−^] also increases from 0.71 to 1.00, and the relative contribution of carbonate in the downstream portion of the GSS outlet dominates completely. A small number of sampling points were distributed around the two sides of the line with the ratio of [Ca^2+^ + Mg^2+^ + Na^+^ + K^+^]*/[HCO_3_^−^] = 1, indicating that the weathering and solution of silicate rock also make an important contribution to solutes in the groundwater of the study area, while sampling point GW05 of SFS showed a higher value of [Na^+^ + K^+^]*/[HCO_3_^−^] because it was close to the distribution area of clastic rock strata. Most of the sampling points were located near the line with the ratio of [Na^+^ + K^+^]*/[HCO_3_^−^] = 0 and [Ca^2+^ + Mg^2+^]*/[HCO_3_^−^] = 1, indicating that the water solute composition in the study area is mainly controlled by carbonate rocks.

#### 5.1.2. Carbonate Dissolution Analysis

The carbonate rock strata in the study area included the lower Triassic strata dominated by limestone and the middle Triassic strata with limestone and dolomite interbedding. To further explore the relative contributions of limestone and dolomite dissolution to the chemical ions of water in the basin, diagrams of Mg^2+^/Ca^2+^ vs. HCO_3_^−^ and Mg^2+^/Ca^2+^ vs. SO_4_^2−^ were drawn. The molar ratio of Mg^2+^/Ca^2+^ can reflect the lithology of the carbonate aquifer through which the groundwater flows. When flowing through the limestone aquifer with calcite as the main mineral, the molar ratio of Mg^2+^/Ca^2+^ was between 0.01 and 0.26, while when flowing through dolomite aquifer, the molar ratio will be greater than 0.85 [44]. As shown in Figure 6, the ratio of Mg^2+^/Ca^2+^ of the samples ranged from 0.058 to 0.82, and the spots of Mg^2+^/Ca^2+^ vs. HCO_3_^−^ were basically distributed in the calcite and calcite-dolomite dissolution area. As shown in Figure 1, the karst system in the north developed in the lower Triassic aquifer dominated by limestone, and the spots were mainly distributed in the calcite dissolution area. The hydrogeological units of FWS, BFS and SFS, which were found to have a longer runoff path and wider recharge area, all contained the middle Triassic aquifer dominated by dolomite, of which the molar ratio of Mg^2+^/Ca^2+^ in groundwater was positively correlated with HCO_3_^−^, which can indicate the water rock interaction intensity. In addition to the process of water rock interaction, the dissolution of dolomite will gradually increase after the dissolution of more soluble calcite, which is the main factor controlling the hydrochemical characteristics of FWS, BFS and SFS. The molar ratios of Mg^2+^/Ca^2+^ showed no obvious correlation with SO_4_^2−^ (Figure 7), indicating that the mineral source in groundwater is not closely correlated with sulfate dissolution.

#### 5.1.3. Analysis of Hydrochemical Characteristics along the Underground River Systems

Analysis of the composition and proportion of the main ions revealed that the karst conduit systems in the north of the Gu Song River were obviously different from those in the FWS, BFS and SFS. Considering the natural separation of the Gu Song River from the karst systems, this section focused on analysis of the hydrochemistry characteristics along the runoff path of the adjacent FWS and north karst conduit systems to further distinguish the differences between them.

The karst conduits in the north are densely developed, with short runoff paths and limited recharge areas. These characteristics result in small-scale development of branches in the north area, and the branches have no obvious impact on the water quality of the main conduits. As shown in Figure 8a, the concentrations of total dissolved solids (TDS), HCO_3_^−^ and other main ions in the north karst conduits are positively related to the runoff distance and negatively related to the elevation. The TDS was found to increase from 190 to 399 mg/L and the SIC from −0.0891 to 0.3024 from the recharge areas to the outlets. The carbonate rock strata along the karst conduits are dominated by limestone; therefore, even though the SID fluctuates, it was not found to be correlated with other elements. There are four karst conduits with a length of more than 2 km in the FSW, and the impact of branches on the water quality of the main conduits cannot be ignored. As shown in Figure 8b, from the recharge area of FWS to sampling point GW13 (elevation of 920 m), the concentrations of TDS, HCO_3_^−^ and other main ions were basically positively correlated with the runoff distance, and the peaks of TDS and HCO_3_^−^ were 358 and 220 mg/L, respectively. These findings were influenced by the water inflows from conduit branches, while TDS and HCO_3_^−^ were reduced to 237 and 149 mg/L, respectively, at the outlet. The elevation of the branch sampling points GW15 and GW16 was 710–720 m, while the TDS and HCO_3_^−^ concentrations of GW15 and GW16 were 139–231 and 88.5–153 mg/L, respectively. The basin of FWS contains limestone and dolomite, and the variations in SIC and SID were found to be similar to those of TDS and HCO_3_^−^: increasing and then decreasing. The gypsum saturation index (SIG) values and fluctuations of the two karst systems were relatively stable, ranging from −1.914 to −2.572 and −1.952 to −2.712, respectively. The features of SIG values and fluctuations, and the lack of an obvious correlation with the molar ratio of Mg^2+^/Ca^2+^ and SO_4_^2−^ supported each other.

### 5.2. Optimized Fuzzy Cluster Analysis

Based on the above discussion regarding differences of hydrochemical and stable isotope data in different karst systems, the fuzzy clustering analysis method was used to classify karst systems quantitatively. Fuzzy cluster analysis is a mathematical method used to investigate and deal with classifications based on the variables of observation samples. This method can classify items that lack reliable historical data, but have similar properties, into one group [45,46]. Fuzzy cluster analysis is widely used in geological exploration, water pollution, pattern recognition and other fields [47]. In traditional fuzzy cluster analysis, it is assumed that each variable has the same control over the results, while in the karst area the weights of different variables such as hydrochemical factors and stable isotopes are obviously different. Based on the characteristics of hydrogeochemical classification in karst area, this study attempted to assign the weight to variables and obtain the optimized fuzzy clustering method to quantitatively analyze the differences between each karst system.

#### 5.2.1. Data Standardization Steps

(1)Analysis steps(1)Data matrixCluster analysis samples constituted sample set $U=\left\{x_{1},x_{2}\cdots ,x_{n}\right\}$, and each sample was represented by m variables: $x_{i}=\left\{x_{i1},x_{i2}\ldots ,x_{\mathrm{im}}\right\}\left(i=1,2,\cdots ,m\right)$.(2)StandardizationDifferent variables may have different dimensions or orders of magnitude. To eliminate the above differences and meet the requirements of the fuzzy matrix analysis, the data were standardized and compressed to the interval of 0 to 1. It is usually necessary to change the translation standard deviation first:(2)$$x_{ik}{}^{\prime }=\frac{x_{ik}-\bar{x_{k}}}{s_{k}}\left(i=1,2,\cdots ,n;\;\;\;\;k=1,2,\cdots ,m;\right)$$
where $\bar{x_{k}}$=$\frac{1}{n}\overset{n}{\underset{i=1}{\sum }}x_{ik}$, $s_{k}=\sqrt{\frac{1}{n}\overset{n}{\underset{i=1}{\sum }}{\left(x_{ik}-\bar{x_{k}}\right)}^{2}}$. After the change, the mean value of variables in the sample was 0, the standard deviation was 1, and the influence of dimension and the order of magnitude between variables was eliminated. The changed $x_{ik}^{\prime }\;$may not be in the interval of 0 to 1, and the translation-range transformation $0\leq x_{ik}^{\prime \prime }\leq 1$ is made, which is suitable for the construction of a fuzzy similar matrix. The translation-range transformation formula is shown in Equation (2), where $x_{ik}^{\prime \prime }$ replaces $x_{ik}$ for clustering analysis.
(3)$$x_{ik}{}^{\prime \prime }=\frac{x_{ik}{}^{\prime }-\underset{1\leq i\leq n}{min}\left\{x_{ik}{}^{\prime }\right\}}{\underset{1\leq i\leq n}{max}\left\{x_{ik}{}^{\prime }\right\}-\underset{1\leq i\leq n}{min}\left\{x_{ik}{}^{\prime }\right\}}\left(k=1,2,\cdots ,m\right)$$(2)Construction of similarity matrixThe traditional clustering method considers the variables in the sample to have the same importance to the sample, and constructs the similarity matrix by calculating the similarity between $x_{i}$ and $x_{j}$. The similarity $r\left(x_{i},\;x_{j}\right)$ is generally characterized by the Euclidean distance method:(4)$$r\left(x_{i},\;x_{j}\right)=\sqrt{\overset{m}{\underset{i=1}{\sum }}{\left(x_{ik}-x_{jk}\right)}^{2}}$$The importance of different variable factors in groundwater in karst areas is obviously different. The weight ($\omega_{k}$) refers to the quantitative allocation of the importance of different indicators of the analysis samples, and the different treatment of the role of each variable in the clustering process. In the process of multi-index evaluation and analysis, $\omega_{k}$ has the function of highlighting key indexes, making the multi-index structure reasonable and realizing the overall optimization. Under the condition of considering the influence of different variables on the similarity degree, the Euclidean distance formula is changed into [48]:(5)$$r\left(x_{i},\;x_{j}\right)=\sqrt{\overset{m}{\underset{i=1}{\sum }}\omega_{k}{\left(x_{ik}-x_{jk}\right)}^{2}}$$(3)Assignment of weightsAt present, the determination of $\omega_{k}$ has entered the stage of combining qualitative and quantitative analysis. By introducing mathematical methods, the theoretical model of determining weights has been developed and improved. There are two kinds of weight determination: subjective and objective [49]. The subjective weighting method is mostly based on the knowledge or experience of experts or individuals, and adopts the qualitative method of comprehensive consulting scoring to determine the weight, and then synthesizes the standardized data. The commonly used methods include the comprehensive index method, the Delphi method, and an analytic hierarchy process [50,51]. The objective weighting method is used to determine the weight according to the correlation between each index or the variation degree of each index value. The objective weight is determined from the data obtained from the survey; therefore, there is no need to consult experts. Common methods of determining this weight include principal component analysis, factor analysis, the variation coefficient method and determination of complex correlation coefficients [52,53].In this study, the Delphi method and AHP method were used to determine the weight of each index. A group of experts in environmental engineering, hydrogeology and other fields was also employed to score the weights of variables in the sample, and the improved AHP method was used for calculation and analysis. The weight assignment steps were as follows:(1)Experts rank the importance of evaluation indicators,${\;x}_{1}>x_{2}\cdots x_{m}$;(2)Experts determine the rational assignment *r_k_* of the ratio of the importance degree of adjacent indexes $x_{k-1}$ and $x_{k}$.
(6)$$r_{k}=\frac{x_{k-1}}{x_{k}}$$
where for $r_{k}=1$, $x_{k}$ and $x_{k-1}$ have the same importance; for $r_{k}=1.2$, $x_{k}$ is slightly more important than $x_{k-1}$; for $r_{k}=1.4$, $x_{k}$ is significantly more important than $x_{k-1}$; for $r_{k}=1.6$, $x_{k}$ is strongly more important than *x_k−_*_1_; where $r_{k}=1.8$, $x_{k}$ is extremely more important than $x_{k-1}$. In this analysis, 1.1, 1.3, 1.5, and 1.7 correspond to the intermediate state of adjacent judgments.According to the rational assignment r determined by the experts, the weight of the m evaluation index $\omega_{m}$ can be calculated as:(7)$$\omega_{m}=\frac{1}{1+\sum_{k=2}^{m}\prod_{i=k}^{m}r_{i}}$$According to the weight $\omega_{m}$, the weight calculation formula of the *m*−1, *m*−2, …3, 2, 1 indicators is as follows:(8)$$\omega_{m}=r_{k}w_{k},\;k=m,m-1,\ldots ..3,2,1$$(4)Output of cluster analysis chartThe fuzzy matrix constructed by the distance obtained from the optimized calculation formula is a fuzzy similarity matrix R. Because it is not necessarily transitive, the intra group connection method is used to achieve its transitivity [54]. In the fuzzy matrix, all samples in the sample set U are classified and gradually merged to form a dynamic clustering diagram.

#### 5.2.2. Sample Analysis

Based on the results of physical survey and hydrogeochemical data analysis, the outlets of the karst system in the north area, FWS, BFS and SFS, and some karst springs distributed in their runoff area were selected as the sample set. The results showed that the solute composition of groundwater in the study area is mainly controlled by water rock interactions, and hardly affected by exogenous materials. The main dissolved minerals were carbonate, followed by silicate. K^+^, Na^+^, Ca^2+^, Mg^2+^, Cl^−^, SO_4_^2−^ and HCO_3_^−^ together account for more than 95% of the total solute contents of groundwater. At the same time, hydrogen and oxygen stable isotopes are important parameters for groundwater recharge, runoff and discharge analysis. Therefore, 10 factors {pH, K^+^, Na^+^, Ca^2+^, Mg^2+^, Cl^−^, SO_4_^2−^, HCO_3_, δD, δ^18^O} were selected as variables for fuzzy cluster analysis. In this study, 20 experts scored the weight of each variable according to the above hydrochemistry and stable isotope data. The improved AHP method was then used to obtain the weight vectors of variables. The average weights of variables were as follows:W=(0.97 0.31 0.022 0.157 0.045 0.047 0.049 0.153 0.152 0.125 0.125)

#### 5.2.3. Results of Fuzzy Cluster Analysis

Figure 9 shows the results of traditional and weighted optimization cluster analysis. Overall, it can be divided into the north karst system in the study area represented by the outlet samples GW03, GW02 and GW06,;the karst systems with long runoff paths and large scales represented by the outlet samples GW25, GW20 and GW14; and rain, which shows the maximum distance from the outlets of each karst system. The samples located in the recharge-runoff area were between rain and outlet points. The samples collected from near the watershed in areas with high elevation had properties similar to those of rain, while the samples collected from near the outlets had properties similar to the outlets. Comparison of the analyses conducted using the two methods revealed that the optimized method produced a classification that is more hierarchical and recognition that is more accurate. Additionally, in the optimization calculation, samples GW21 and GW18 were identified and belonged to the runoff area of karst area represented by GW25, GW20 and GW14, and they are obviously different from other samples belonging to the runoff area of karst area represented by GW03, GW02 and GW06. These findings objectively reflect the characteristics of the groundwater cycle process.

Table 5 shows the Euclidean distances of water samples. The optimized Euclidean distance was 0.039–0.267, and the maximum distance was observed in GW03 and the rain sample. In addition to rain, the maximum distance of groundwater samples was 0.255, which indicates the difference of the recharge and runoff area between BFS represented by GW17 and north karst systems represented by GW05. Additionally, GW03, GW02 and GW06 represent the north karst systems and have a Euclidean distance of 0.042–0.068, while GW25, GW20 and GW14 represent SFS, BFS and FWS, respectively, and have a Euclidean distance of 0.043–0.056. The distance among the north karst systems, FWS, BFS and SFS, was 0.086–0.118. These findings are supported by the results of geophysical survey and hydrogeochemical data analysis.

### 5.3. Groundwater Environmental Risk Zone Identification

This study defined the karst conduits’ development area and recharge-runoff area of the drinking groundwater source together as a risk zone. In the risk zone, human activities that may produce pollution, such as chemical industry, mining, centralized livestock and poultry farming will be restricted.

The results of geophysical survey, hydrochemical data and cluster analysis clearly show that FWS and BFS differ from the north karst systems. However, no clear boundary was found between conduit 6 and 7 or conduit 13 and 7. Therefore, in view of the safety of the drinking groundwater source, the risk zone was designated with the recharge-runoff area of FWS and BFS; conduit 6 and conduit 14 are the boundaries, giving an area of 33.90 km^2^ and accounting for 34.5% of the study area (Figure 10). In the risk zone, the industrial activities that may have an impact on the groundwater environment should be prohibited or restricted.

## 6. Conclusions

This study carried out a detailed investigation based on the objective of drinking groundwater source protection in typical karst areas in the southwest of China. Geophysical survey, hydrogeological chemical process analysis and optimized fuzzy cluster analysis were applied to conduct a systematic study on the development of karst conduits. Each step employed in this study was mutually dependent and corroborated. Results of each step gradually enhanced the understanding of karst system, and effectively improved the rationality and accuracy of drinking groundwater source risk zone designation of karst aquifer. Finally, 34.5% of the study area was designated as risk zone. In the risk zone, controlled by the high gradient between 0.04–0.23 of karst conduits system, once the pollutants enter the groundwater system, they will quickly migrate to the drinking groundwater source. There is not enough response time to carry out remediation to prevent the drinking groundwater source from being polluted. Therefore, in order to avoid the human health risk caused by the drinking groundwater sources pollution, results of this study can be submitted to the local government. The result will be an important scientific basis for the construction of local regulations on the protection of drinking groundwater sources. The regulations need to specify that human activities, which may produce pollutants or have a negative impact on the groundwater environment, shall be prohibited or restricted in the risk zone.

## Figures and Tables

**Figure 1 ijerph-17-03627-f001:**
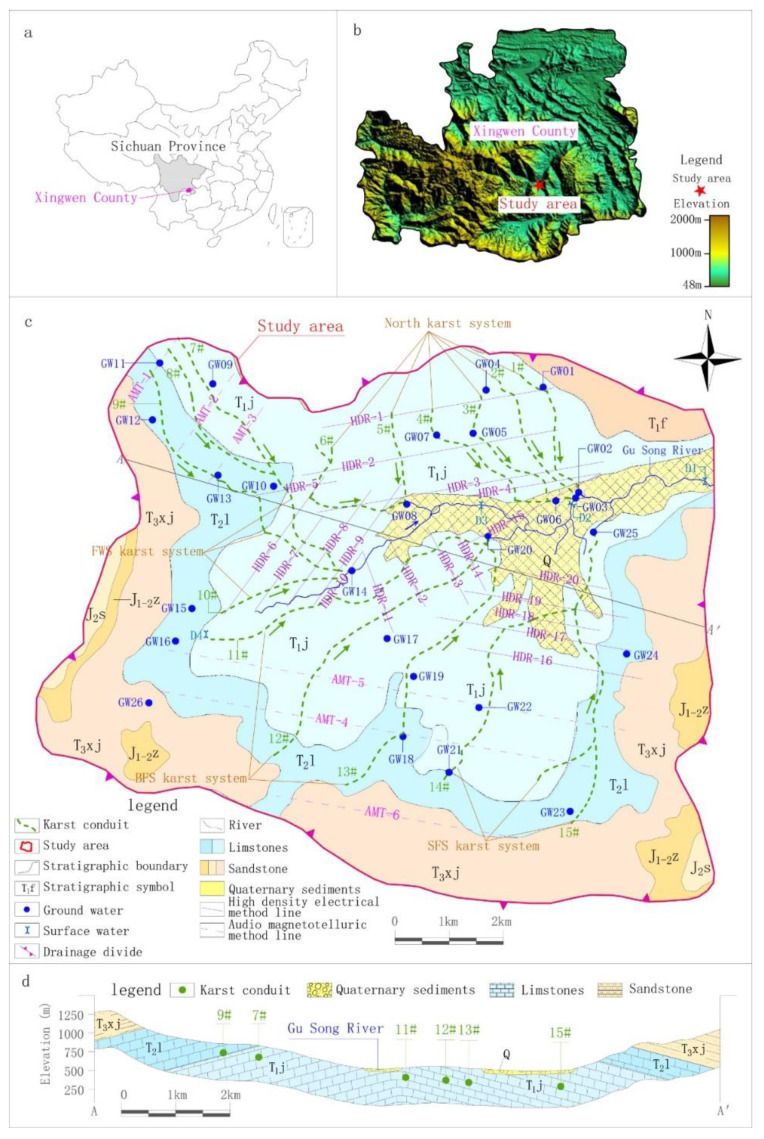
Location, sampling sites and hydrogeological map of the study area. (**a**) Location of the study area, (**b**) Topographic map of the study area. (**c**) Sampling sites and hydrogeological map of the study area, (**d**) Hydrogeological profile map of the study area.

**Figure 2 ijerph-17-03627-f002:**
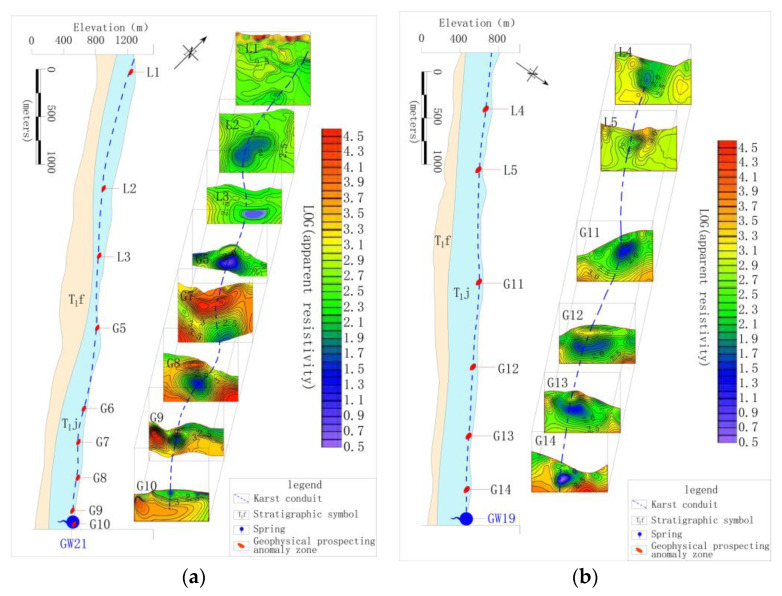
(**a**) Interpretation of geophysical survey section of 9 karst conduit; (**b**) interpretation of geophysical survey section of a 12karst conduit.

**Figure 3 ijerph-17-03627-f003:**
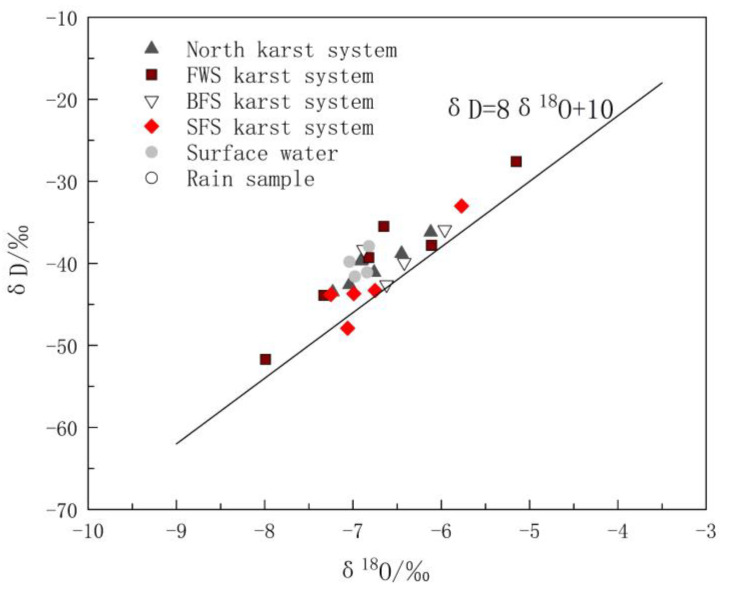
Hydrogen and oxygen isotopic compositions of water samples in the study area.

**Figure 4 ijerph-17-03627-f004:**
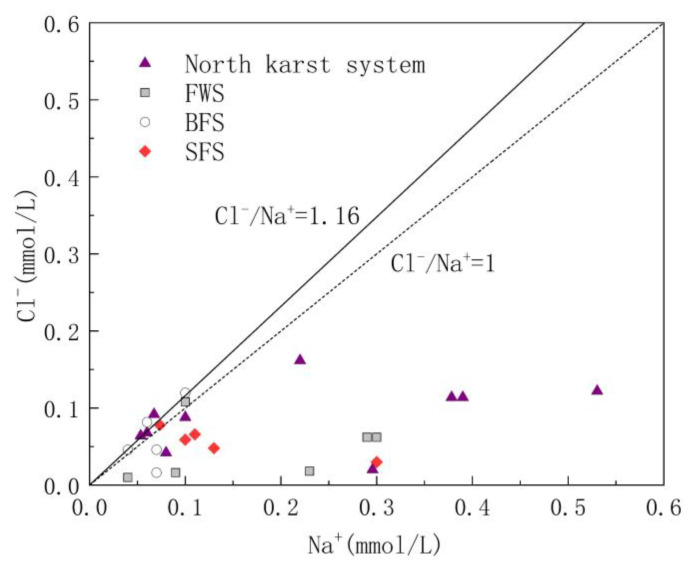
Relationship diagram of Cl^−^/Na^+^ in underground water samples.

**Figure 5 ijerph-17-03627-f005:**
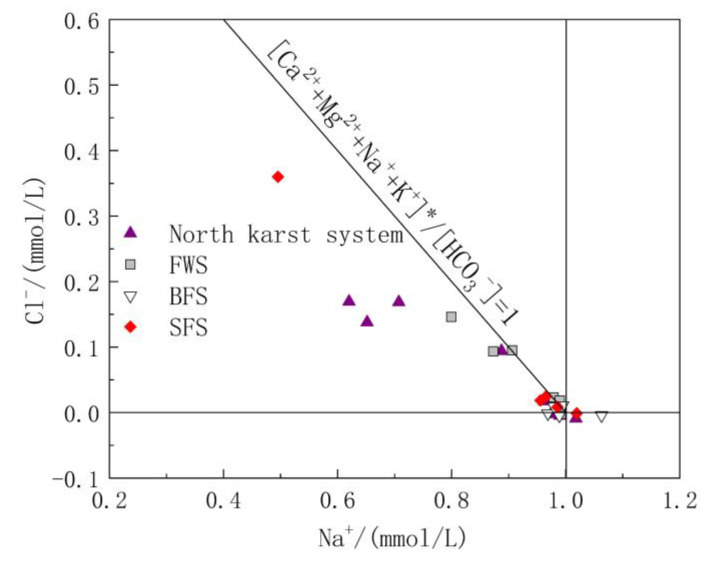
Relative contribution of weathering of carbonate and silicate rocks.

**Figure 6 ijerph-17-03627-f006:**
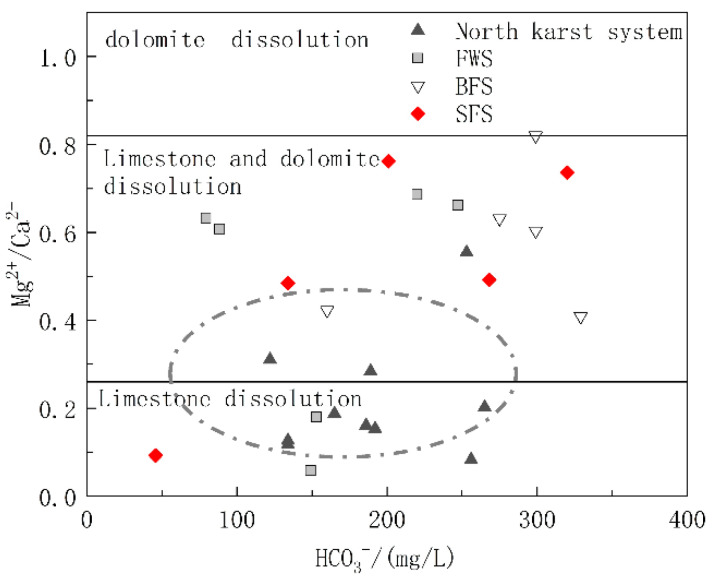
Relationship of Mg^2+^/Ca^2+^ and HCO_3_^−^ in underground water samples.

**Figure 7 ijerph-17-03627-f007:**
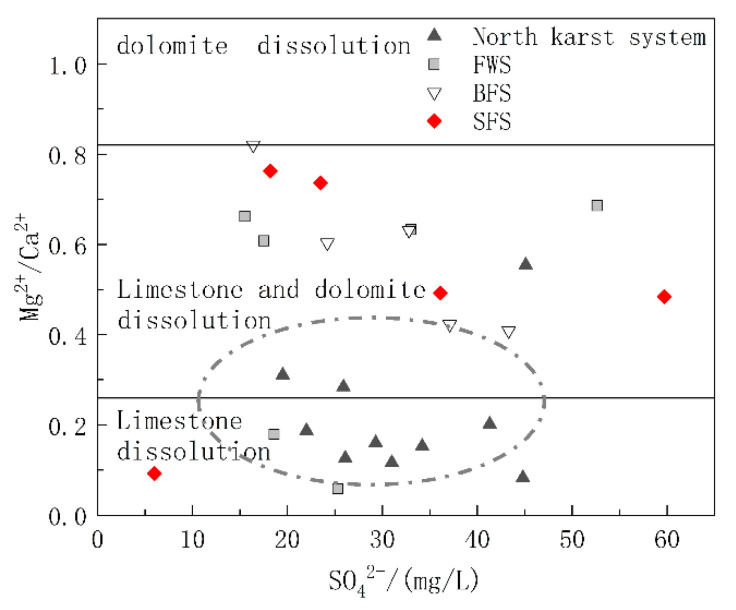
Relationship of Mg^2+^/Ca^2+^ and SO_4_^2−^ in underground water samples.

**Figure 8 ijerph-17-03627-f008:**
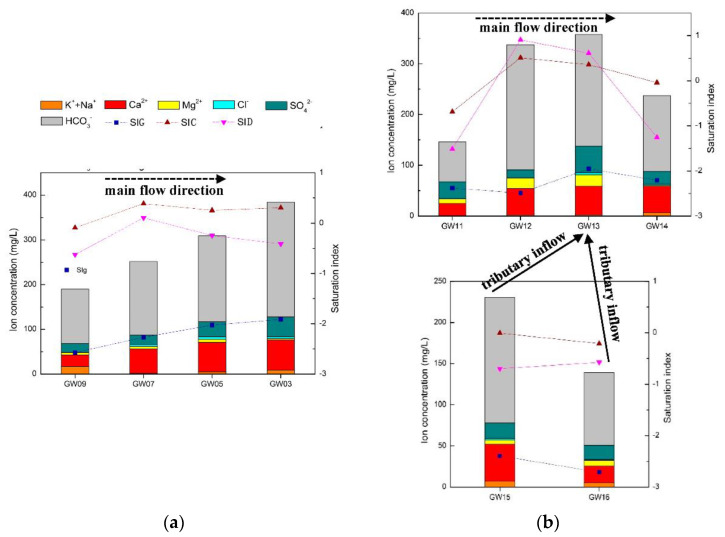
(**a**) North karst system hydrochemistry variation along the main conduit; (**b**) FWS karst system hydrochemistry variation along the main conduit and the influence of tributaries.

**Figure 9 ijerph-17-03627-f009:**
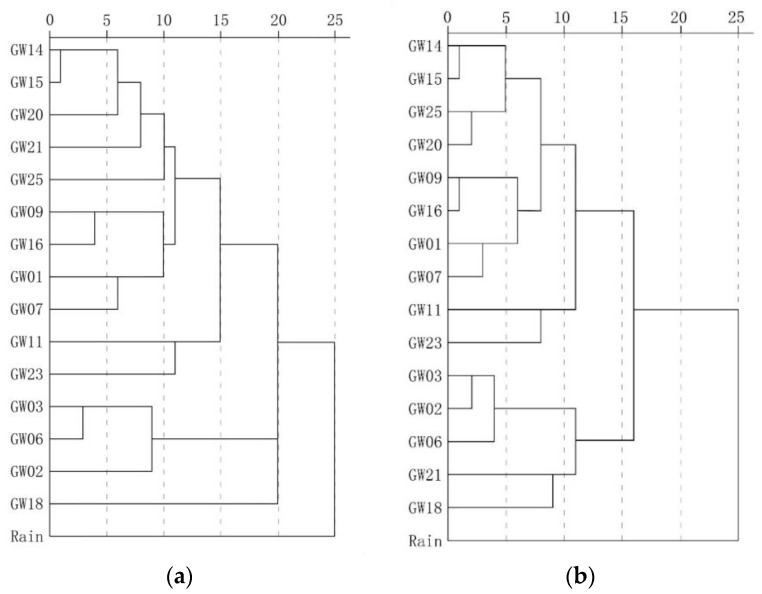
Comparison of the results of traditional clustering analysis and optimization clustering analysis. (**a**) Traditional fuzzy clustering analysis. (**b**) Optimized fuzzy clustering analysis.

**Figure 10 ijerph-17-03627-f010:**
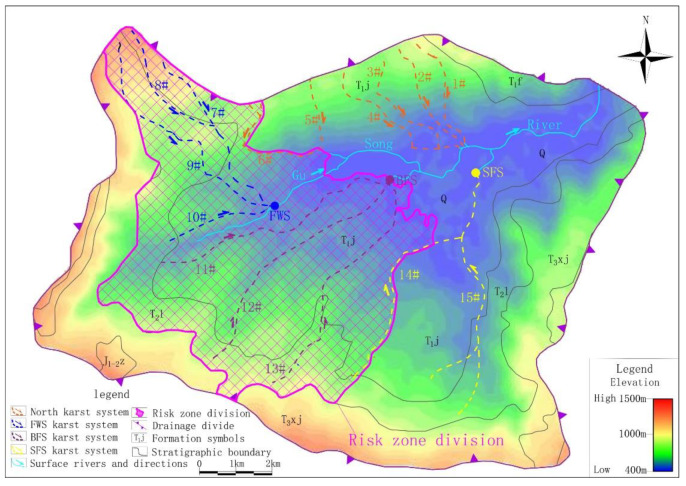
Results of risk zone designation.

**Table 1 ijerph-17-03627-t001:** Sampling point basic information.

Sample Point No.	Stratum	Location Information	Elevation (m)
GW01	T_1_j	Water-filled cave along 1 conduit	960
GW02	T_1_j	Outlet of XZS (1 conduit)	455
GW03	T_1_j	Outlet of LXS (2 conduit)	454
GW04	T_1_j	Spring near 3 conduit	880
GW05	T_1_j	Water-filled cave along 3 conduit	745
GW06	T_1_j	Outlet of GSS (3 and 4 conduit)	459
GW07	T_1_j	Spring along 7 conduit	840
GW08	T_1_j	Outlet of BCS (5 and 6 conduit)	514
GW09	T_1_j	Spring near 7 conduit	1235
GW10	T_2_l	Spring near 7 conduit	1000
GW11	T_2_l	Spring near 9 conduit	1330
GW12	T_2_l	Spring near 9 conduit	1220
GW13	T_2_l	Water-filled cave along 9 conduit	920
GW14	T_1_j	Outlet of FWS (7, 8, 9 and 10 conduit)	528
GW15	T_2_l	Spring near 10 conduit	720
GW16	T_2_l	Spring near 11 conduit	710
GW17	T_1_j	Spring near 12 conduit	670
GW18	T_2_l	Water-filled cave along 13 conduit	870
GW19	T_1_j	Water-filled cave near 13 conduit	800
GW20	T_1_j	Outlet of BFS (11, 12 and 13 conduit)	475
GW21	T_1_j	Spring near 14 conduit	820
GW22	T_1_j	Spring near 14 conduit	680
GW23	T_2_l	Water-filled cave near 15 conduit	835
GW24	T_2_l	Spring near 15 conduit	670
GW25	T_1_j	Out of SFS(14 and 15 conduit)	475
GW26	T_3_xj	Spring near 11 conduit	920
D1	-	Surface water section downstream of north karst system outlets	435
D2	-	surface water section downstream of BFS, SFS and FWS outlets	450
D3	-	Surface water section downstream of FWS outlet	480
D4	-	Surface water section upstream of FWS outlet	650

**Table 2 ijerph-17-03627-t002:** Analytical methods and minimum detection values.

Number	Monitoring Factors	Groundwater		
		Test Method	Instrument	Minimum Detectable Value (mg/L)
1	K^+^	Flame atomic absorption spectrometry	AA-700	0.03
2	Na^+^			0.01
3	Fe			0.03
4	Mn			0.01
5	Cu	Atomic absorption spectrometry	ICE-3500	0.001
6	Zn			0.05
7	Cd	Graphite furnace atomic absorption method	ICE-3500	0.0001
8	Pb			0.001
9	Cl^−^	Ion chromatography	ICS-600	0.007
10	SO_4_^2−^			0.018
e	NO3-			0.016
12	NO2-			0.016
13	Volatile phenols	4-AAP spectrophotometric	SP-721E	0.0003
14	NH4+	Nessler’s Reagent Spectrophotometry		0.025
15	Cr6+	Spectrophotometric method of dibenzoyl dihydrazide		0.004
16	Al	The Chromazurol S Spectrophotometric Method		0.008
17	Hg	Atomic fluorescence spectrometry	AFS-933	0.00034
18	As			0.0003
19	Ca^2+^	Flame atomic absorption spectrometry	AA-700	0.02
20	Mg^2+^			0.002
21	HCO_3_^−^	titration	\	5
22	Total hardness	Titration of disodium ethylenediamine tetraacetic acid	\	1
23	COD_Mn_	Acid potassium permanganate titration	\	0.05
24	Petroleum	Infrared spectrophotometry	OIL480	0.01
25	pH	Glass electrode method	PHSJ-4F	\
26	TDS	Weighing method	ZA220.R4	0.05

**Table 3 ijerph-17-03627-t003:** Details of karst conduits.

Name	Conduit No.	Initial Elevation (m)	Outlet Elevation (m)	Possible Conduit Length (km)	Possible Average Hydraulic Gradient	Possible Buried Depth of Conduit (m)		Mean Flow (L/s)	
						Runoff Area	Adjacent to Discharge Area	Dry Season	Wet Season
XZS	1	945	455	2.72	0.18	65~107	21~41	3.8	9.2
LXS	2	943	454	2.64	0.19	73~158	16~23	4.4	10.2
GSS	3	857	459	2.73	0.15	119~140	17~34	40.5	101.2
	4	851		3.40	0.12	70~134	20~42		
BCS	5	881	514	1.69	0.16	71~135	27~46	17.4	45.3
	6	967		3.01	0.15	96~150	21~27		
FWS	7	1322	528	5.72	0.14	82~188	33~47	325.3	813.2
	8	1274		3.19	0.23	62~209	28~47		
	9	1221		7.21	0.10	63~139	18~48		
	10	737		2.19	0.10	92~104	17~35		
BFS	11	658	475	2.19	0.08	85~154	19~48	529.3	1215.0
	12	887		5.71	0.07	50~181	31~36		
	13	847		5.68	0.07	55~197	22~47		
SFS	14	840	459	5.73	0.07	46~188	32~37	418.6	1045.2
	15	636		4.52	0.04	57~175	33~40		

**Table 4 ijerph-17-03627-t004:** Hydrochemical characteristics of groundwater.

No.	Monitoring Factors		The Dry Season			The Wet Season		
			Detection Rate	Conc. Range	Average Conc.	Detection Rate	Conc. Range	Average Conc.
1	pH		100%	7.02~8.46	7.62	100%	7.15~8.25	7.71
2	Total hardness		100%	55~240	161.94	100%	30~330	188
3	TDS		100%	71~276	178.56	100%	71~504	303
4	Medal	Fe	56.7%	3.10 × 10^−2^~6.80 × 10^−2^	4.5 × 10^−2^	56.7%	2.70 × 10^−2^~5.30 × 10^−2^	4.50 × 10^−2^
		Cd	46.7%	1.90 × 10^−5^~1.00 × 10^−4^	3.00 × 10^−5^	40.0%	1.00 × 10^−5^~1.00 × 10^−4^	2.00 × 10^−5^
		Al	100%	8.00 × 10^−3^~9.80 × 10^−2^	2.20 × 10^−2^	100%	1.00 × 10^−2^~1.45 × 10^−1^	5.27 × 10^−2^
5	COD_Mn_		100%	0.76~1.67	1.15	100%	0.69~1.49	1.38
6	Nitrogen compounds	NO_3_^−^	100%	3.11~12.3	5.68	100%	4.17~14.37	8.43
		NO_2_^−^	6.7%	0.27	0.27	0%	-	-
		NH_4_^+^	46.7%	0.03~0.13	0.06	60%	0.017~0.19	0.06
7	Aqua Chem	K^+^	100%	0.262~10.6	1.15	100%	0.92~28.7	4.94
		Na^+^	100%	0.342~5.01	2.40			
		Ca^2+^	100%	7.8~58.6	43.59	100%	11~94.2	53.10
		Mg^2+^	100%	1.14~27.9	10.64	100%	0.61~29.2	12.78
		Cl^−^	100%	1.33~9.31	3.53	100%	0.35~5.74	2.39
		SO_4_^2−^	100%	19.1~82.8	35.53	100%	6~59.7	29.96
		HCO_3_^−^	100%	43~207	137	100%	45.8~329	198.60

**Table 5 ijerph-17-03627-t005:** Euclidean distance of optimization algorithm.

Sample	GW03	GW02	GW06	GW25	GW20	GW14	GW01	GW09	GW11	GW15	GW16	GW07	GW23	GW21	GW18	Rain
GW03	0.000	0.050	0.056	0.086	0.085	0.118	0.131	0.149	0.170	0.118	0.168	0.125	0.211	0.099	0.098	0.262
GW02	0.050	0.000	0.076	0.091	0.087	0.117	0.136	0.135	0.165	0.115	0.161	0.134	0.196	0.099	0.117	0.248
GW06	0.056	0.076	0.000	0.109	0.097	0.106	0.134	0.157	0.182	0.118	0.183	0.128	0.211	0.128	0.144	0.267
GW25	0.086	0.091	0.109	0.000	0.051	0.076	0.073	0.083	0.103	0.073	0.095	0.085	0.147	0.086	0.131	0.209
GW20	0.085	0.087	0.097	0.051	0.000	0.063	0.090	0.093	0.112	0.064	0.111	0.107	0.133	0.075	0.140	0.197
GW14	0.118	0.117	0.106	0.076	0.063	0.000	0.059	0.072	0.116	0.039	0.100	0.089	0.120	0.112	0.176	0.185
GW01	0.131	0.136	0.134	0.073	0.090	0.059	0.000	0.058	0.100	0.071	0.067	0.056	0.133	0.128	0.172	0.204
GW09	0.149	0.135	0.157	0.083	0.093	0.072	0.058	0.000	0.092	0.067	0.043	0.096	0.103	0.119	0.180	0.169
GW11	0.170	0.165	0.182	0.103	0.112	0.116	0.100	0.092	0.000	0.128	0.076	0.096	0.092	0.163	0.215	0.227
GW15	0.118	0.115	0.118	0.073	0.064	0.039	0.071	0.067	0.128	0.000	0.098	0.108	0.125	0.084	0.157	0.159
GW16	0.168	0.161	0.183	0.095	0.111	0.100	0.067	0.043	0.076	0.098	0.000	0.097	0.103	0.137	0.192	0.182
GW07	0.125	0.134	0.128	0.085	0.107	0.089	0.056	0.096	0.096	0.108	0.097	0.000	0.153	0.153	0.179	0.251
GW23	0.211	0.196	0.211	0.147	0.133	0.120	0.133	0.103	0.092	0.125	0.103	0.153	0.000	0.176	0.255	0.168
GW21	0.099	0.099	0.128	0.086	0.075	0.112	0.128	0.119	0.163	0.084	0.137	0.153	0.176	0.000	0.095	0.180
GW18	0.098	0.117	0.144	0.131	0.140	0.176	0.172	0.180	0.215	0.157	0.192	0.179	0.255	0.095	0.000	0.265
Rain	0.262	0.248	0.267	0.209	0.197	0.185	0.204	0.169	0.227	0.159	0.182	0.251	0.168	0.180	0.265	0.000

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
