# Peer review of "Application of Geophysical and Hydrogeochemical Methods to the Protection of Drinking Groundwater in Karst Regions"

_ijerph, 2020, doi:10.3390/ijerph17103627_

Round 1

Reviewer 1 Report

Title: Application of geophysical and hydrochemical methods to the protection of drinking groundwater in karst regions

Authors:  Kai Song, Guangxu Yang, Fei Wang, Jian Liu, Dan Liu

General remarks:

The manuscript concerns application of geological survey and hydrochemical analysis in groundwater protection study. The authors at the end propose an area of protection to preserve the local drinking water resources.

The authors proposed an interesting case study, with an important amount of data. The manuscript is worthy of being published but needs some serious modification before publication.

In general, I recommend to use a classical plan: Introduction / Study Area / Material and Methods / Results / Discussion / Conclusion. In the present version there is no “Results” section, all is given in discussion part. Besides, the discussion and conclusion should be enlarged.

Most of the figure labels should be enlarged and the notations in equation and in the text for isotopes must be carefully homogenized.

I made several specific comments in the rest of the document.

Specific comments to the paper:

Line 11: “development of karst pipelines” do you mean speleogenesis processes? I'm a little bit confused about what you mean by "karst pipelines" here. Is it human build pipeline in karst area or is it natural development of cave/conduits by rock dissolution? Maybe karst conduit is more appropriated here.

Line 13: may cause pollution to drinking water quality drinking water pollution

Line 14:” physical prospecting” you mean geophysical survey?

Line 18: “in the north” can you provide the study area before (country or region)

Line 18-31: You provided here so much results for an abstract, it would be better to focus on the main implication of your research and the methods used in your work.

Line 33:” the difference of different between karst water systems

Line 33-38: it appears quite unclear what means “groundwater drinking water source risk zones”.

Line 55:” source for more than 15,000 people in the study area of this paper.” Even if the “study area” might be presented before please give at least country (or region/city) concern by de drinking water plan.

Line 56: The introduction sentence for geophysical survey introduction would benefit for a larger definition, such as hydrogeophysics. Maybe you can add the reference for Chalikakis et aL. 2011

Chalikakis K, Plagnes V, Guerin R, et al (2011) Contribution of geophysical methods to karst-system exploration: an overview. Hydrogeol J 19:1169. https://doi.org/10.1007/s10040-011-0746-x

Line 73: I think the sentence might be reformulated, then the references appears as numbers [23-25] and the text would be easier for the reader.

Line 78: “development characteristics, water quality characteristics and formation and variation mechanisms in a study area dominated by karst aquifers”. It sounds unclear what is formation and variation mechanism. Please reformulate this sentence.

Line 83:” development characteristics of karst pipelines”

Line 92: No need for GPS coordinates in the text

Line 94: does 101.68 km2 corresponds to the recharge area for the Gusong river at sampling site SW01 (figure 1)? In the figure we see the Limestone surrounded by Sandstone. So, the question is does your “study area” corresponds in fact to the “recharge area”?

Line 125: “Sweden software RES2DINV was applied to conduct used to perform precise inversion of two-dimensional resistivity of the detection results.“ What do you mean by “precise inversion” ? By definition inversion can’t be “precise” but you can estimate the quality of fitness between simulated data and field measurement through a RMSE or NRMSE coefficient. Also, can you add information about the norm used for the inversion (robust or not ?) Can you also add information about the error on you inversion model.

Line 142:” regarding complex karst development characteristics.”

Line 143:” In the study, Samples were taken during the dry season”

Line 148:” and limit of for each factor.”

Line 149: This paragraph would benefit of relevant literature dealing with the used methods

Line 158: The equation must be numbered + be careful about notation for isotopic ratio in both equation and in the text. The notation should be homogeneous in all manuscript.

Line 160: The table 1 should not be separated on 2 pages, if needed please use an entire page

Line 161: The section must be renamed “Results”

Line 174:” horizontal circulation zonation” is it the vadose zone? You can go to the paper from Jouves et al 2017 where you can find a sketch about vertical and horizontal drainage structures according the different compartment of the aquifer.

Jouves J, Viseur S, Arfib B, et al (2017) Speleogenesis, geometry, and topology of caves: A quantitative study of 3D karst conduits. Geomorphology 298:86–106. https://doi.org/10.1016/j.geomorph.2017.09.019

Line 225: Please add a reference here corresponding to the Shoka Lev classification

Line 228: “the strong hydraulic alternation condition” it is unclear, please reformulate de the sentence

Line 232: It would be easier to read if you use scientific format for the concentration values.

Line 331: “These feature […] supported each other.” This sentence is a bit unclear, maybe just delete because it does not provide information.

Line 373: There are 4 levels of sections opened at the same point. It would be better to provide a clearer overview to the reader. Also, there are problems in section numbers in the all 4.3.1. section. This part should be reformulated for a better clarity in the different step of data processes.

Line 377: I don’t think it is necessary to go so far in the detail for cluster analysis. It would be better to short this theoretical part and provide some relevant literature about cluster analysis.

Line 382: “interval [0,1].” Please use another notation since this one is for bibliography references. Same problem in line 387.

Line 474: The all table 4 should be on the same page.

Line 475: This section should be renamed “Discussion”. There are lot of results on each part of your work (geophysical survey, hydrochemical analysis and the proposed clustering approach). Nonetheless, there are a lack of data confrontation and discussion about the general objectives of your studies. The figure 10 is not cited in the text. Maybe you can provide more discussion about the following points:

  • What is the implication of the “Risk zone division”? For example can you provide an estimation of transit time if contaminant are rejected on surface or a volume of water impacted?
  • Can you quantify or propose a way to quantify the uncertainty about the definition of the “risk zone division”? Is this one corresponds to favorable or unfavorable consideration? Is it possible to provide several level of security for the contour of the zone?

Specific comments to the figure:

Figure 1: The figure is well presented but some elements appear unclear:

  • “Karst pipeline”: are they inferred from geophysical survey or speleological investigation?
  • “Stratigraphic boundary”: as there is no change in color for formation T3xj, J1-2z, J2s the grey line can be confused with altitude line. So, it would be better to change the color for those formations
  • The arrow next to the karst pipeline symbol are generally used for fault sense of motion. This might correspond to sense of water flow, so please change the notation on the map.
  • The notation between HDR and AMD profile is unclear and the names or not easy to read on the map.

Figure 7:

  • What are the continuous black arrows? I suppose it’s tributary but please complete the legend.
  • Why is there line between GW15 and GW16 as it is not considered in the flow direction as for the other graphs? Then the figurative should be different from continuous line.
  • As it is the same color for both (a) and (b), there is no need for the legend to be twice in the figure 7

Figure 10:

  • “Drainage divide” should also present in figure 1
  • I think you should inverse color scale à hot color for high elevation
  • You can remove “Karst Pipeline” next to the reference on the map
  • The figure can be improved by using a more appropriated legend.
  • If you are using graticules on the map, please add also on figure 1 or remove here. Also, if you keep graticules it should be in the same format as in the text (line 92)

Reviewer 2 Report

The manuscript named " Application of geophysical and hydrogeochemical methods to the protection of drinking groundwater in karst regions " written by Kai Song and co-workers provide theoretical support for the protection of centralized drinking water sources in karst areas. This work provides a scientific basis for the accurate designation of risk zones that may cause pollution to drinking water quality. Also, physical prospecting, hydrogeological chemical process analysis and optimized fuzzy cluster analysis were used to gradually improve the understanding of karst water systems. The topic the study is interesting and the manuscript is well structured. The overall quality of the study is suitable to be published in Chemosphere. The manuscript can be accepted for publication after minor revisions.

(1) Some figs. Should be improved.

(2) Line 383-450. References should be cited.

Reviewer 3 Report

Paper name:

- paper name is correctly formulated.

Abstract:

- in 430 words, describes the paper content in the desired format, but even here it is clearly visible how extensive is the presented work and that that paper should perhaps be divided into two or three parts – containing geophysical investigations, geochemistry and results of fuzzy cluster analysis. Terminologically very misleading term of “karst pipeline” for the first time appears here. It should be replaced by “karst conduit system” / “cavity system” / “system of karstic voids” everywhere in the text. Term “water-filling karst pipelines” is also terminologically, but also scientifically incorrect – karstic voids and cavities are water-filled, but also many times filled by clay material with substantially lowered electric resistivity, so one cannot directly recognize water/clay filling without visiting the whole system. In line 14, physical prospecting should be as → geophysical prospecting. Abbreviations as “XZS, LXS, GSS and BCS, FWS, BFS and SFS” karst systems are inappropriate in the abstract – they are explained in the text, but abstract should be presented independently. Perhaps only “northern and southern karst systems” should be mentioned instead.

Key words:

- in keywords, karst pipeline → karst conduits systems; groundwater drinking water sourse → drinking groundwater sources;  /  ; (stream water; groundwater; Kaidu River Basin; hydrochemical process; water-rock interaction; human activity) are adequately selected;

Text:

- line 51: studies of “pore type” are irrelevant to karst hydrogeology, perhaps studies of heterogeneity / karstification should be more appropriate;

- lines 113-114: spring flow was between 0.1 L/s and 1215.0 L/s represents very general information, it should be explained whether these are mean (average discharges, or absolute minimum of all springs and absolute maximum of all springs);

- lines 125-126: Sweden software → Swedish software;

- line 142: the sentence starts with “Hydrochemical and isotopic data contained important information”, but it should be known to the reader that these are hydrochemical and isotopic data on water – water is not mentioned here!

- line 150: it should be somehow explained why the stable isotopes were added as monitoring factors only during the wet season (and not the dry one);

- line 161: why the text is completely missing the chapter results and directly jumps into Discussion?

- line 172: carbonatite distribution area → carbonate rocks area;

- line 176: karstic voids and cavities are many times filled not only by water, but also by clay material with substantially lowered electric resistivity, so merely by geophysical prospection we cannot directly recognize water/clay filling of the conduit system;

- lines 192-204: karstification is a difficult process, where many flow / conduit branches may appear – is this characteristic as given in these lines really so clear concerning lengths and gradients, or the authors may “relativize” those statements by using the words such as “possibly” / “probably” / “perhaps”;

- lines 199-204: please explain whether discharges given for individual karstic conduit systems are average values, or only results of one-time measurements;

- line 212: please explain what is the meaning of “optimal sampling” mentioned here;

- line 225: classification of Shoka Lev should be also cited and referenced!

- lines 216-251: although the name of the chapter is “Basic characteristics of groundwater quality and chemistry”, rain and surface water is mentioned directly at its beginning, and later it remains unclear to the reader  whether characteristics given here are for all samples (how many of them) or only for groundwater samples (how many of them)…

- line 235: CODMn is a parameter that has no concentration at all, “CODMn value” should better be used instead;

- line 275: section D4 to D1 is mentions, but the reader should be informed where are these sections – on some figure, perhaps;

- line 296: although it is claimed that “As shown in Figure 1, the karst system in the north developed in…”, these facts are not shown in Figure 1;

- line 324: inflow of branchs → water inflows from conduit branches?

- lines 334-360: signs of “‰” are completely missing in the isotope characteristics, and surely should be supplemented;

- line 338: again, as isotopic composition of rainfall / precipitation in whole is the most variating in all water cycle, it should be better explained why only one rainfall sample was taken;

- lines 357-360: this is a nice part and perhaps good example of different circulation, but the readers are absolutely not familiar with the GW5 spring, and some characteristic of all sampling places (in a table, perhaps) should be definitely incorporated in the paper;

- lines 374-380: the way of presenting optimized fuzzy cluster analysis for karst groundwater systems work progress is more typical for a report than for a scientific paper;

- lines 453-455: again, the readers are absolutely not familiar with the GW-type samples mentioned here and any characteristic of all sampling places is missing in the paper;

- lines 484-493: text in these lines is very general and should better appear in the introduction, it seems that the whole (long) abstract appear here in the Conclusions, while text from the lines 493-503 after some corrections, are more suitable to be abstract.

Figures:

- Figure 1: this figure is referred in the text also as the source of information on local geology, but legend is obviously connected only with the cross-section beneath the main figure, and also the stratigraphic symbols given in the map are not explained as limestones / sandstones / dolomites etc…. This should be inevitably improved, or perhaps another (geological map) should be added).

- Figures 3-6: these figures are not too readable although when deeply zoomed in, some enhancement should be performed perhaps;

- Figure 10: is a nice figure demonstrating the conclusions, but absolutely not mentioned in the text;

Tables:

- Table 1: as horizontal lines are missing, the relevant test method for monitored parameter is sometimes unclear.

Round 2

Reviewer 1 Report

Dear authors,

thanks for the corrections on your manuscript. It brings real added value to your work.

There are still a few mistakes and typos :

  • line 63 : references from Zhu et al. Sharma et al. and Chetelat et al. are not cited in accordance with the editor format (numbering for the references)
  • line 226 : mg/l vs mg/L

I recommand a moderate English changes, if possible with native english speaking because some sentences still poorly formulated and there are some typos.

Sincerly

Reviewer 3 Report

After corrections realised and documented by the authors, the paper can be published.